# Oral Supplementation of Specific Collagen Peptides Combined with Calf-Strengthening Exercises Enhances Function and Reduces Pain in Achilles Tendinopathy Patients

**DOI:** 10.3390/nu11010076

**Published:** 2019-01-02

**Authors:** Stephan F.E. Praet, Craig R. Purdam, Marijke Welvaert, Nicole Vlahovich, Gregg Lovell, Louise M. Burke, Jamie E. Gaida, Silvia Manzanero, David Hughes, Gordon Waddington

**Affiliations:** 1Department of Sport Medicine, Australian Institute of Sport, Leverrier St, Bruce ACT 2617, Australia; marijke.welvaert@canberra.edu.au (M.W.); nicole.vlahovich@ausport.gov.au (N.V.); greglovell544@gmail.com (G.L.); silvia.manzanero@gmail.com (S.M.); david.hughes@ausport.gov.au (D.H.); gordon.waddington@canberra.edu.au (G.W.); 2University of Canberra Research Institute for Sport and Exercise (UCRISE), Cnr Allawoona St & Ginninderra Drive Bruce, ACT 2617, Australia; jamie.gaida@canberra.edu.au; 3Department of Physiotherapy, Australian Institute of Sport, Leverrier St, Bruce ACT 2617, Australia; cpurdam@gmail.com; 4Department of Sports Nutrition, Australian Institute of Sport, Leverrier St, Bruce ACT 2617, Australia; louise.burke@ausport.gov.au; 5Discipline of Physiotherapy, University of Canberra, Building 1/11 Kirinari St, Bruce ACT 2617, Australia

**Keywords:** achilles tendon, microvessels, contrast-enhanced ultrasound, hydrolysed collagen supplementation

## Abstract

The current pilot study investigates whether oral supplementation of specific collagen peptides improves symptoms and tendon vascularisation in patients with chronic mid-portion Achilles tendinopathy in combination with structured exercise. Participants were given a placebo or specific collagen peptides (TENDOFORTE^®^) in combination with a bi-daily calf-strengthening program for 6 months. Group AB received specific collagen peptides for the first 3 months before crossing over to placebo. Group BA received placebo first before crossing over to specific collagen peptides. At baseline (T1), 3 (T2) and 6 (T3) months, Victorian Institute of Sports Assessment–Achilles (VISA-A) questionnaires and microvascularity measurements through contrast-enhanced ultrasound were obtained in 20 patients. Linear mixed modeling statistics showed that after 3 months, VISA-A increased significantly for group AB with 12.6 (9.7; 15.5), while in group BA VISA-A increased only by 5.3 (2.3; 8.3) points. After crossing over group AB and BA showed subsequently a significant increase in VISA-A of, respectively, 5.9 (2.8; 9.0) and 17.7 (14.6; 20.7). No adverse advents were reported. Microvascularity decreased in both groups to a similar extent and was moderately associated with VISA-A (*R*_c_^2^:0.68). We conclude that oral supplementation of specific collagen peptides may accelerate the clinical benefits of a well-structured calf-strengthening and return-to-running program in Achilles tendinopathy patients.

## 1. Introduction

The treatment of chronic Achilles tendinopathy remains challenging with a relatively high percentage of non-responders. There are many proposed treatment options, of which an eccentric exercise program is currently the first treatment of choice. Despite the good results on pain scores after eccentric exercises, a 5-year follow-up study showed that only ~40% of the patients were completely pain free and 48% had received one or more alternative treatments [1]. In accordance with this, there is still a need for treatments that would improve the benefits of a structured eccentric exercise program. Recent studies indicate that both cardiometabolic [2,3], as well as nutritional factors [4,5] can modulate local tendon healing. Although clinical studies are still scarce, a recent International Olympic Committee (IOC) consensus statement on dietary supplements in high-performance athletes [6] proposes that dietary supplementation containing gelatin or hydrolysed collagen could potentially be useful for athletic populations as increased intake of collagen-derived peptides has been shown to modulate collagen synthesis [7] and reduce tendon- [8] and joint-related pain [9,10]. Others have shown that glycine, as the most abundant component of collagen hydrolysates [11], has disease-modifying properties in both animal [12,13] and in vitro [14] models of tendinopathy. There is also first evidence, that collagen peptides might improve functional ankle properties in chronic ankle instability [15]. Therefore, we aimed to study the potential clinical benefits of a specific hydrolysed collagen supplement as add-on therapy to 6 months of an eccentric calf-strengthening and well-structured return-to-running program.

It has been well established that increased (micro) vascularity is associated with chronic (painful) tendon lesions [16]. As a change in (micro)vascularity might be a hinge for improved tendon architecture, our aim was to check whether the oral intake of collagen peptides influence microvascular changes in tendinopathic areas of the Achilles tendon using real-time contrast-enhanced ultrasonography (CEUS) [17]. In a recently published study [18], it was shown that CEUS provides a more objective and reproducible assessment of tendon microvascularity in comparison with power Doppler ultrasound. Furthermore, CEUS was also shown to be moderately associated with increased Achilles tendon symptoms. We hypothesized that improvements in tendon pain and function throughout the course of the intervention would be reflected by a reduction in CEUS-based tendon microvascularity. A previous ultrasonographic tissue characterisation study has shown that structural integrity is decreased in both Achilles tendons in people with unilateral Achilles tendinopathy [19]. As dietary supplementation of hydrolysed collagen appears to have systemic effects on collagen-dense tissue [20,21,22], we aimed to study both tendons of each individual patient, independent of whether participants had uni- or bilateral symptoms.

## 2. Methods

The current study is a prospective double-blinded placebo-controlled clinical trial with a cross-over design. Through balanced within block-randomization (computer-based, 5 individuals per block), eligible participants were allocated to each study arm by an independent staff member to either 3 months of bi-daily 2.5 g hydrolysed specific collagen peptides (sCP) (TENDOFORTE^®^, GELITA AG) or a placebo product to be dissolved in water, both pre-packed in identical sachets (see also study flow diagram Figure 1). The bi-daily intake of sCPs was based on the pharmacokinetic profile; the placebo sachets contained 2.5 g of maltodextrin (CARGILL) obtained by enzymatic conversion of starch. The sachets containing sCP or placebo were absolutely identical in appearance and the products were equal in flavor and texture. After 3 months, participants who received the sCPs were crossed-over to the placebo product (group AB) and vice versa (group BA). Allocation concealment was further ascertained by allocating an individual to group AB or group BA. No patient, research nurse, assessor or any other medical or research staff was aware of the treatment assignments for the duration of the study. All data and statistical analyses were also completed before the randomization code was broken at the end of the completed trial [23]. Based on lack of previous data sets that allowed for a power analysis and sample size calculation, we designed a small-scale pilot study with multiple outcome measures that involved 2 groups of 10 participants. Data collection of this single-center trial was performed in an outpatient sports medicine clinic of a national high-performance sports institute. All subjects gave their informed consent for inclusion before they participated in the study. The study was conducted in accordance with the Declaration of Helsinki, and the protocol was approved by the local ethics committee of the Australian Institute of Sport (Project ID: 20141201V2) and prospectively registered (ANZCTR#12615001035516). Patients were not involved in our study design. Participants were recruited by social media and advertisements. In- and exclusion criteria are summarized in Appendix A
Table A1.

### 2.1. Exercise Intervention

In addition to the nutritional intervention, all participants followed a well-structured eccentric calf-strengthening program and milestone-based return-to-running program for a total duration of 6 months. The eccentric exercise started with a calf raise using the unaffected limb (or both limbs if the symptoms were bilateral), followed by an eccentric drop using the injured leg. The exercise was performed with both the knee straight and the knee bent to target the gastrocnemius and soleus muscles respectively. Over a 6-month period, participants were instructed to perform 2 × 90 repetitions daily, despite the presence of pain [24]. All participants were instructed to avoid weight-bearing sporting activities for the first four weeks. When participants reported less than 2 out of 10 pain on single leg hopping, they were allowed to start with low-intensity running exercises. Details and milestones used within this program can be found in Appendix A
Table A2. Half an hour before the calf-strengthening exercises, participants were instructed to ingest 2.5 g of sCP or placebo product dissolved in a glass of cold water. This provides sufficient time for orally ingested sCPs to be absorbed in the gut and appear in the blood stream [25]. Participants kept a diary of their calf-strengthening exercises and were asked to return any leftover sachets at both their 3 and 6 months follow-up appointments.

### 2.2. Outcome Measurements.

All participants underwent a series of measurements and evaluations at baseline, 3 and 6 months to evaluate their response to the combined exercise and nutritional intervention. All measurements and questionnaires were scheduled within a time frame of 3 days and performed by the same team of health practitioners and researchers who were blinded for the nutritional intervention and trained in the standardized operating procedures for the different measurements. Both at 3 and 6 months, our participants were asked whether they thought they had been taking the sachets with sCPs or placebo. At the end of the trial and after deblinding this information was matched with true allocation order.

### 2.3. Victorian Institute of Sports Assessment–Achilles (VISA-A) Questionnaire, Patient Satisfaction and Return-to-Running

One day prior to the ultrasound examinations, all patients were asked to rate their subjective Achilles tendon pain and functional limitations for both the right and left Achilles tendon separately using a Victorian Institute of Sports Assessment–Achilles (VISA-A) questionnaire [26]. After 3 and 6 months patients also received a questionnaire to rate their satisfaction and to what level they had been able to resume their running sports activities (Appendix A
Table A3).

### 2.4. Real-Time Harmonic Contrast Enhanced Ultrasound (CEUS) Measurements

Contrast-enhanced ultrasound (CEUS) measurements scans were performed as described in previous studies [17,18]. Patients were prone with fully extended knees and both ankles immobilized with an ankle angle between 5–10 degrees plantar flexion whilst ensuring minimal tendon tension. Both Achilles tendons were examined in the longitudinal plane with a 58 mm long high-frequency (5–14 MHz) linear array transducer (model 14L5/PLT-1005BT) on an Aplio™500 (TUS-A500, Toshiba/CANON Medical Systems, Tochigi Otawara, Japan). The transducer was longitudinally positioned in the Achilles tendon midline using a retort clamp attached to the bed. A 12G copper wire was placed between transducer, gel and skin to locate and mark tendon insertion in a reproducible manner. Tendon insertion was defined as the attachment point of most ventral tendon fibres to the calcaneal bone. A total of seven skin marks at 1-cm intervals were used to position the ultrasound probe over the tendon mid-portion (2–6 cm). Probe pressure was kept to a minimum to avoid blood vessel obliteration. Scanning parameters were optimised for 3 cm depth and kept constant for all participants. Real-time harmonic CEUS of the Achilles tendon mid-portion was performed through the injection of 10uL.kg^-1^ body weight bolus of agitated contrast agent (DEFINITY™, Lantheus Medical Imaging, Billerica North Billerica, MA, USA) in a forearm vein, followed by 10 ml 0.9% saline solution. CEUS is based on second harmonic imaging of intravascular circulating microbubbles with a diameter of ~1–3 microns. These microbubbles act as echo-enhancers and are used with the aim to identify small vessels (down to 40 microns) [18]. Participants were instructed not to talk or change body position during recordings. Blood pressure and heart rate were measured every 5 min throughout the procedure with a digital sphygmamometer and automatically inflating arm cuff (OMRON HEM-7130, Wujia Medical Equipment Co., Ltd., Fengshan District, Kaohsiung City, Taiwan). As peak flow of CEUS signal was achieved within 2 minutes following the bolus injection, only first 120 s of digital CEUS data were used for further analysis. To allow for complete washout of perflutren gas through the lungs, a minimum of 15 min was allowed between consecutive CEUS measurements of right and left Achilles tendon. Dedicated contrast harmonic image quantification software (2B771-005EN*V, Toshiba/CANON Medical Systems, Tochigi Otawara, Japan) was used for time-curve analysis (TCA). In the present study, it involved the examination of a 25 × 13 mm region of interest (ROI) that included at least 2 mm of the anterior fat pad. Within this ROI, the relative microvascularity (MV) values were determined as peak log-scaled signal intensity in decibels (dB) of a normalised smoothed curve during the first 2 min after subtraction of baseline signal intensity 20–25 s after bolus injection. Inter-observer reliability for CEUS-based MV-values was previously shown to be excellent with an intraclass correlation (ICC) of 0.97 [17].

### 2.5. Blood Sampling

At baseline, 3 and 6 months, non-fasting blood samples were obtained from venipuncture to determine total cholesterol, triglycerides and uric acid and exclude (familial) hypercholesterolaemia [27] and gout [28] as a contributing factor to tendinopathy. In addition, at each time point a 500 µL serum sample was stored at −80 °C for further analyses in case of a serious adverse event related to the intake of the investigational product.

### 2.6. Product Safety and Monitoring of Adverse Events

Specific collagen peptides are characterized by a considerably high safety profile. So far no clinical indications of allergies have been observed. No incompatibilities with other diets or medications have ever been described in the medical literature. Collagen peptides received the GRAS status from the Food and Drug Administration. “GRAS” is an official abbreviation which stands for “generally recognized as safe [29]. Nevertheless, participants were instructed to report any gastro-intestinal or other side effects that they thought were potentially associated with the intake of the investigational product or placebo.

### 2.7. Statistical Analyses

All data are presented as means ± standard deviation (SD), unless indicated otherwise. Baseline values of all parameters were tested for uniformity via discriminant analysis. All data processing and statistical analyses were performed before de-blinding the randomization order. Linear mixed model (LMM) analysis using R statistical software package was used to determine the association between CEUS-based microvascularity values and VISA-A scores at the 3 different time points. For statistically significant interactions, we determined *R*_c_^2^, which represents the variance explained by the combination of fixed effects and random effects. Mixed analysis of variance (ANOVA) was used for all other comparisons. LMM analysis was also performed to test for changes in our main outcome variables between examination at baseline and following the 3- and 6-month intervention periods between and within groups in a cross-over design. The LMM analysis satisfies all conditions of an intention-to-treat analysis and has also been shown to be the most powerful statistical method under a variety of conditions that are common to randomized controlled trials [30]. Carry-over effects within our cross-over design were accounted for in our LMM-analyses by looking at interaction of the order and treatment effect across time. A *p* value < 0.05 was considered statistically significant.

## 3. Results

### 3.1. Study Population

From September 2015 until April 2016, a total of 60 patients (32 men/28 women, age 43 ± 10 years) with subjective Achilles tendon symptoms showed interest in our study. Following initial screening, 40 potential candidates (18 men/22 women, age 43 ± 11 years) were excluded or withdrew for the reasons mentioned in Figure 1. A total of 20 patients (13 men/7 women, age: 44 ± 8 years, body mass index (BMI): 24.4 ± 3.3 kg∙m^−2^) with clinical symptoms (mean duration 54 ± 90 months) of uni- (*n* = 7) or bilateral (*n* = 13) mid-portion Achilles tendinopathy were formally included in the study following a detailed medical history and physical examination by a sport and exercise physician (Figure 1). Demographic characteristics and blood test results of participants at baseline are presented in Table 1. To evaluate homogeneity of the data at baseline between the 2 groups a discriminant analysis was carried out. The data revealed no statistically significant differences between both study groups. All participants wished to return to pain-free running and had either uni- or bilateral Achilles tendon tenderness on palpation. The clinical diagnosis was based on the presence of tenderness and/or tendon thickening 2 to 6 cm proximal to the distal insertion. One asymptomatic tendon that previously had undergone surgery for a complete Achilles tendon rupture was excluded from our analysis.

### 3.2. Change in VISA-A Scores

Changes in Achilles tendon pain and function scores during the treatment were evaluated through the VISA-A questionnaire [26]. Baseline VISA-A scores from right and left limb were not significantly correlated (Pearson’s R: 0.39 (95% confidence interval (CI): −0.08; 0.72), *p* = 0.097). VISA-A score for the group AB at T1 was on average 60.8 (95%CI: (52.0–69.6)), similar to the group BA at T1 (62.8, 95%CI: (54.0–71.6)). Linear mixed modeling statistics showed that after 3 months, estimated mean difference in VISA-A score for group AB was 12.6 (95%CI: (9.7–15.5)) points, while in group BA this was 5.3 (95%CI (2.3; 8.3)) points (Figure 2). After crossing over the AB and BA group showed an estimated mean difference in VISA-A score of respectively 5.9 (95%CI (2.8–9.0) and 17.7 (95%CI (14.6–20.7) between T2 and T3. Both group AB and BA achieved the same estimated mean improvement in VISA-A score between T1 and T3 of respectively 18.5 (95%CI: (15.5–21.6)) and 23.0 (95%CI: (19.9–26.1)) (Figure 2). Importantly, our LMM analyses showed that there was a statistically significant interaction from our sCP randomization order ((LMM, F (2,762) = 20.3, *p* < 0.0001)) on change in VISA-A scores indicating a significant difference between the groups in evolution of the VISA-A scores over time in relation to sCP versus placebo supplementation.

### 3.3. Patient Satisfaction and Return-to-Running Sports

After 3 months respectively 6 out of 10 participants in group AB and 7 out of 10 participants in group BA reported their treatment as ‘good or excellent’. After 6 months 7 out of 9 participants in group AB and 7 out of 9 in group BA reported being satisfied with their treatment. The success of the intervention is also reflected in Figure 3, summarizing the number of participants that were able to return to their desired (running) sport, although none of them was able to return to their pre-injury level within the duration of the study. None of the participants in group AB reported that they stopped running sport activities after switching over from sCPs to placebo.

### 3.4. Change in Achilles Tendon Microvascularity and Blood Markers

Both group AB and BA showed a significant (LMM, F (2,202) = 12.8, *p* < 0.001) decrease in tendon microvascularity over the course of the intervention as assessed by CEUS. As is illustrated in Figure 4, there is no difference in effect between the 2 interventions and no evidence for a benefit of sCP supplementation on tendon vascularisation. The difference at T2 for the 2 different randomisation groups is only marginally significant (LMM, F (2,202) = 2.81, *p* = 0.062), due to the greater variability in group AB compared to the BA group, in particular on that time point. Our LMM analyses indicate a significant interaction between microvascularity and VISA-A symptom score over the course of the study (*R*_c_^2^: 0.68; LMM, F (2,196) = 3.32, *p* = 0.038). Non-fasting serum levels of mean total cholesterol, triglycerides and uric acid were unchanged at T2 and T3 (Table 2, ANOVA, *p* > 0.05)

### 3.5. Compliance with Interventions and Adverse Events

In group AB, no compliance data on the daily calf strengthening exercise were available for 1 participant at both T1-T2 and T2-T3. In group BA, compliance data on the daily calf strengthening exercise were missing for 1 participant (T1-T2), 1 participant (T2-T3), and 1 participant (T1-T2 and T2-T3). Based on the available exercise diaries, both groups AB and BA showed and achieved a similar compliance of respectively 84 ± 11% and 78 ± 14% (*p* = 0.326) to the twice-daily calf-strengthening exercises over the 6 months intervention period.

In group AB, no compliance data regarding intake of investigational product were available for 1 participant at both T1-T2 and T2-T3. In group BA, compliance data were missing for 3 participants (T1-T2) and 1 participant (T2-T3). Compliance to intake of sCPs and/or placebo was respectively 89 ± 15% and 91 ± 9% (*p* = 0.906) for group AB and BA over the 6-month intervention period. At T2, 6 out of 10 in group AB and 4 out of 10 in group BA correctly guessed their allocation to active or placebo supplement. At T3, 4 out of 9 in group AB and 5 out of 9 in group BA correctly guessed their allocation to active or placebo supplement. A total of 1 participant in group BA reported minor gastro-intestinal discomfort after ingestion of the investigational product during the first week of the treatment whilst using the placebo. Following a medical check-up with an independent physician who was deblinded for the randomization order of this participant, it was decided that there was no clear association with the dietary intervention and this participant was advised to continue with the trial. After 1 week, a medical follow-up learned that all gastro-intestinal symptoms had disappeared. Except for mild delayed onset muscle soreness of the calf muscles in response to the eccentric exercise, no other adverse events were reported during the course of the intervention.

## 4. Discussion

The main finding of this small-scale but well-controlled pilot study is that oral supplementation of sCPs with high glycine content may accelerate the clinical benefits of a well-structured calf-strengthening and return-to-running program in patients with uni- or bilateral chronic Achilles tendinopathy symptoms. Although the study was not powered to detect differences in VISA-A scores or return-to-running sport activities between the 2 groups, it is interesting to note that, independent of randomization order, both clinical outcome variables appear to improve more during the sCPs supplementation period. Although the differences between the 2 groups at 3 months may appear small, the mean change in VISA-A score in group AB of 12.6 (9.7; 15.5) points is well above the minimum clinically important difference (MCID) of 6.5 for Achilles tendinopathy patients [31]. In our placebo group BA, VISA-A increased only by 5.3 (2.3; 8.3) points after 3 months, which is below the MCID of 6.5. After crossing over from sCP to placebo and vice versa, group AB and BA showed an opposite response with, subsequently, a significant increase in VISA-A of respectively 5.9 (2.8; 9.0) and 17.7 (14.6; 20.7). These clinical findings extend on two earlier studies [32,33], that reported respectively a pain modulating and in vitro anti-inflammatory effect of a nutraceutical containing collagen, mucopolysaccharides and vitamin C. Furthermore, in a rodent model of collagenase-induced Achilles tendinopathy, a 5% glycine-rich diet for 3 weeks improved hydroxyproline, glyosaminoglycans and non-collagenous protein content of the Achilles tendon [12]. Glycine has also been shown to improve collagen matrix organisation strength and tenocyte remodelling, most likely by modulating both TNF-alpha, matrix metalloproteases and the availability of collagen precursors [14]. As the sCP supplement in the present study contains 22% of glycine [11], it could be postulated that the increased intake of 1.1 g of glycine per day may have contributed to observed clinical improvements in VISA-A scores during sCP supplementation. Due to nature of the co-intervention, a true washout period to bring participants back to their baseline condition is in Achilles tendinopathy patients not possible. From a pharmacokinetic perspective, the serum hydroxyproline content following the ingestion of hydrolysed collagen peptides returns to baseline within 12 h [25]. Nevertheless, any potential carry-over effects have been accounted for in our LMM-analyses by looking at interaction of the order and treatment effect across time.

Given the small scale of the study and unequal distribution of men and women in our 2 study groups, there is a potential of selection bias. Although the adaptability of tendon to loading differs in men and women [34], we are not aware of any intervention study that specifically showed clinically relevant gender difference in VISA-A scores following a calf-strengthening program. To evaluate homogeneity of the data at baseline between the 2 groups a discriminant analysis was carried out. The data revealed no statistically significant differences between both study groups. Unfortunately, the relatively low number of female participants does not allow for a gender-based inference. As women may be more prone to develop Achilles tendinopathy [35,36], future random-controlled trials (RCTs) should aim to include an equal number of men and women in both study arms.

In order to maximize any potential synergistic effect of orally ingested sCPs with bi-daily calf-strengthening exercises (as per Alfredson’s protocol) on tendon collagen synthesis, our participants were instructed to ingest the sCPs 30 min before each exercise session. About 15–30 min after oral ingestion of collagen hydrolysates, free and peptide forms of serum hydroxyproline levels significantly increase and reach a maximum after 30–60 min and are almost back to baseline level after 7–12 h [25,37]. Although this was not investigated as a separate outcome measure, the small time window between the bi-daily intake of a sachet with 2.5 g of sCPs and the calf-strengthening exercise may have contributed to the high compliance rate observed in our trial. A recent proof-of-concept study by Shaw et al. (2017) showed a dose-dependent improvement of collagen metabolism in ligaments by the oral administration of gelatine 60 min before jumping exercises [7]. This specific time window was based on the fact that peak availability of free and peptide forms of serum hydroxyproline is respectively 1 and 2 h following the oral ingestion of gelatin [38]. Based on the new insight that the stimulation of exercise-induced anabolic processes might be influenced positively by the oral administration of sCPs immediately before an exercise program [7], we recommend that future study designs should take absorption kinetics of the available forms of gelatin or sCPs into consideration.

A strength of the present study is that there were only 2 dropouts after T2, 1 in each group, which is a negligible number. Given that our LMM approach operates on a missing at-random-assumption, we deemed the reasons for dropout not violating the assumptions of the model. Furthermore, attrition is low and no individuals were excluded from our LMM so the possibility of selection bias is considered low in the present study. Nevertheless, as the clinical improvements in VISA-A scores after 3 months are in the same range as previously published studies on eccentric calf-strengthening programs [39], no firm conclusions can be drawn regarding the added benefits of sCP supplementation in chronic Achilles tendinopathy. So, although the present randomized controlled and double-blinded crossover study found a significant interaction of sCP supplementation on change in VISA-A score during a 6-month calf-strengthening plus structured return-to-running program, our findings require duplication in a larger and adequately powered clinical trial.

From a clinical perspective, it is important to note that the majority of our participants had a relatively long history of mid-portion Achilles tendinopathy not responding to rest or other physiotherapy regimes. The latter is also reflected by the fact that placebo group BA only showed a clinically relevant improvement in VISA-A score after they were switched over to sCPs supplementation. This cross-over effect indicates that sCPs supplementation could be a useful adjunct therapy in Achilles tendinopathy patients whom are not responding to a well-standardized eccentric calf strengthening exercise program.

To improve our understanding of the in vivo working mechanism of specific collagen peptides, we also investigated the microvascular changes of the Achilles tendon using a novel medical imaging technique based on contrast-enhanced ultrasonography. The present study is the first of its kind showing that objectively quantified Achilles tendon microvascularity is inversely and moderately associated with Achilles tendon symptoms throughout the course of a therapeutic intervention. The latter indicates that CEUS of tendons may be a useful and more objective medical imaging technique to monitor the clinical response to a therapeutic intervention. Although a previous publication showed that CEUS has a better sensitivity in detecting microvascular abnormalities as compared to power Doppler ultrasound (PDU) [18], the observed change in CEUS-based microvascularity during the course of the intervention was not associated with the intake of specific collagen peptides. Our rather large inter-subject variability in Achilles tendon microvascularity at baseline may have precluded detecting any statistically significant differences between the 2 groups. Nevertheless, our results are well in line with several other PDU-based imaging studies that reported rather equivocal responses in microvascularity following conservative treatment of Achilles tendinopathy [40,41,42,43,44]. Despite the longitudinally observed association between CEUS-based microvascularity and Achilles tendon symptoms, it is unlikely that CEUS will be useful in determining the potential in vivo working mechanism of specific collagen peptides as an adjunct treatment to a well-structured eccentric calf strengthening exercise program.

Eccentric tendon loading has shown to be a safe and effective method to reduce pain and to improve tendon structure [45]. Although PDU-based studies have been equivocal [40,41,42,43,44], eccentric tendon loading has been shown to reduce the number of neovessels in the tendinopathic area of the tendon matrix. The latter is considered to be an important etiological mechanism for the beneficial outcome of a calf strengthening program as also applied in the current study [46]. Cell-line studies have shown that tenocytes produce the antiangiogenic factor endostatin, a proteolytic fragment of Collagen XVIII, in response to physiological mechanical load, thus limiting neo-angiogenesis [47]. Although endostatin response to eccentric loading has not been investigated in diseased tendons, several mechanistic studies [47,48,49] indicate that only the right amount of tissue loading increases endostatin expression and, as such, may reduce microvascularity as observed in the present study. As the collagen matrix of tendon is a highly mechanosensitive tissue, cytokine homeostasis and cell survival underlie an intimate balance between adequate biomechanical stimuli and disturbance through load deprivation and overload. This delicate balance between tendon blood flow and loading pattern was recently highlighted in a study by showing that the risk of developing Achilles tendinopathy increased if blood flow increase in the tendon after running was reduced [36]. Based on these new insights, future studies on novel therapeutic strategies for tendinopathy should preferably assess tendon blood flow response both at rest as well as following a standardized tendon-loading protocol.

Although from a pathophysiological perspective return-to-running status and tendon microvascularity are likely to interact, the limited statistical power of the present pilot study does not allow us to exclude such a significant interaction using linear mixed modelling. The decision to return to running was a function of pain reduction during single leg hopping as well as the successful and pain-free achievement of pre-defined milestones during the course of our rehabilitation program. The latter can be justified by the well-established fact that progressive (and sports-specific) tendon loading is required for successful tendon healing and return to sport [38]. Furthermore, the milestones described in Appendix A
Table A2 entailed significantly lower Achilles tendon loading than present during pre-injury sports participation. Although the VISA-A questionnaires are designed to quantify both activities-of-daily-living (ADL) and running-associated Achilles tendon pain, we cannot exclude that running with incompletely healed tendon pathology may have introduced a certain level of bias towards the overall VISA-A score. As such, for future RCTs we would recommend assessing both ADL-dependent and running sport-dependent Achilles tendon pain on a visual analog scale as secondary outcome measures.

Previous in vitro work on fibroblasts indicates that biosynthesis of ligament and tendon matrix molecules, as well as elastin content can be stimulated by exposure to sCPs [50]. A recently published shear wave elastography study indicates that, compared to controls, participants with Achilles tendinopathy display lower Achilles tendon elastic modulus [51]. Accordingly, we recommend that future randomized clinical trials in Achilles tendinopathy patients should include shear wave elastography to monitor changes in tendon elasticity following the intake of sCPs.

## 5. Conclusions

Although this pilot study has limited statistical power and requires duplication in a larger clinical trial, oral supplementation of sCPs may accelerate the clinical benefits of a well-structured calf-strengthening and return-to-running program in patients with chronic Achilles tendinopathy symptoms. Further imaging studies are required to understand the potential in vivo therapeutic working mechanism of sCPs within tendons.

## Figures and Tables

**Figure 1 nutrients-11-00076-f001:**
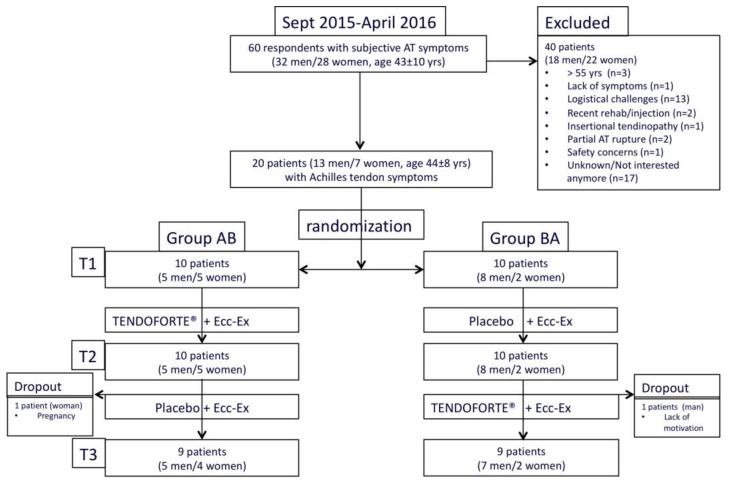
CONSORT study flow diagram.

**Figure 2 nutrients-11-00076-f002:**
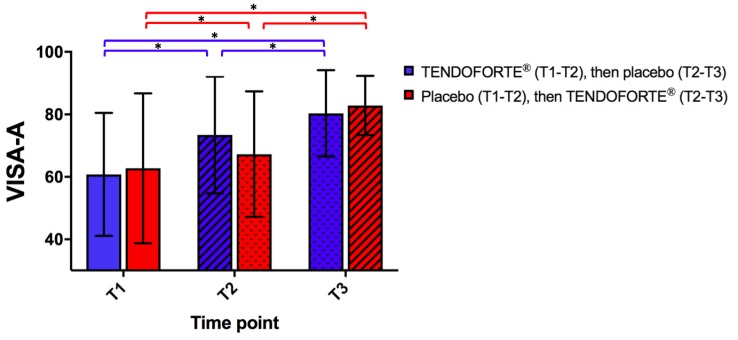
Progression of the mean VISA-A scores during the course of the study Data represent mean ± SD. Statistical analysis of changes in VISA-A scores at baseline (T1), 3 months (T2) and 6 months (T3) via linear mixed modeling. *p* values ≤ 0.05 were considered statistically significant and marked with an asterisk.

**Figure 3 nutrients-11-00076-f003:**
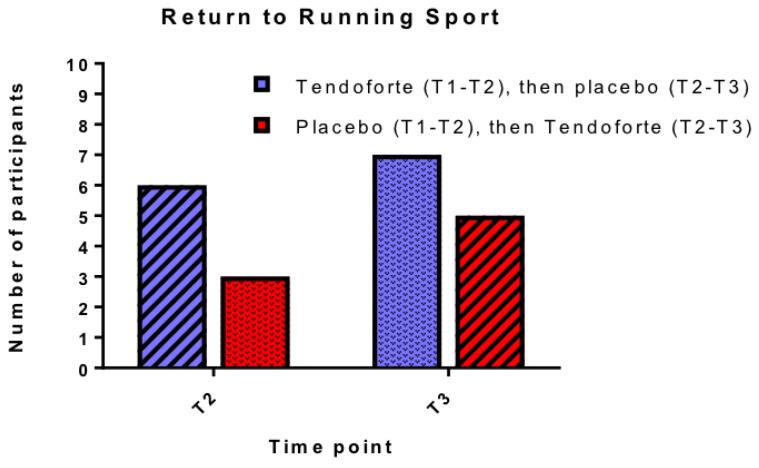
Number of participants able to return to their running sport at T2 and T3. Due to low numbers Chi-square statistics were not possible on these data.

**Figure 4 nutrients-11-00076-f004:**
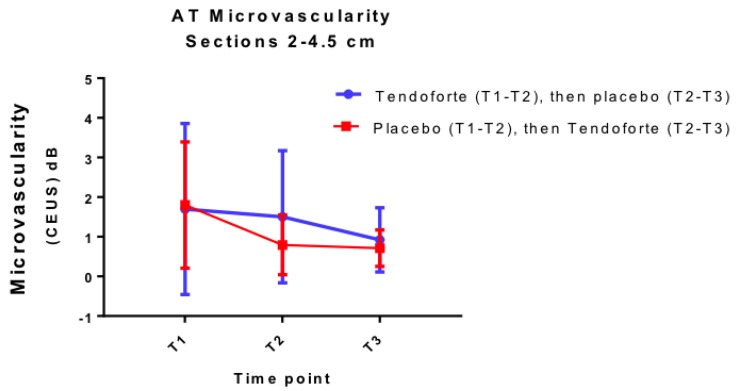
Changes in microvascularity of the Achilles tendon mid-portion (2–4.5 cm from insertion) over time (both symptomatic and asymptomatic tendons combined).

**Table 1 nutrients-11-00076-t001:** Baseline-characteristics study population (*n* = 20).

	Group AB	Group BA	Total
Total	10	10	20
Male/Female	5/5	8/2	13/7
Age (years) *	45.3 ± 6.4	42.0 ± 9.4	43.7 ± 8.0
Body mass index (BMI) (kg.m^−2^) *	23.4 ± 3.23	25.5 ± 3.3	24.4 ± 3.3
Total Cholesterol (mmol∙L^−1^)	5.6 ± 0.7	5.2 ± 0.8	5.4 ± 0.8
Triglycerides (mmol∙L^−1^)	1.9 ± 2.1	1.7 ± 1.1	1.8 ± 1.7
Urate (mmol∙L^−1^)	0.3 ± 0.1	0.4 ± 0.1	0.4 ± 0.1
Symptom duration (months) ^&,#^	24 (0–360)	32 (0–180)	18 (0–360)
VISA-A _symptomatic tendons_ *	58.5 ± 19.9 (*n* = 17)	55.9 ± 21.9 (*n* = 15)	57.6 ± 20.3 (*n* = 32)
VISA-A _asymptomatic tendons_ *	77.0 ± 18.4 (*n* = 3)	88.5 ± 10.3 (*n* = 4)	84.7 ± 12.9 (*n* = 7)

VISA-A: Victorian Institute of Sports Assessment–Achilles, * mean ± SD, ^&^ median, minimum-maximum, ^&^ based on pooled data of both symptomatic and asymptomatic tendons, # all tendons.

**Table 2 nutrients-11-00076-t002:** Serum blood makers at baseline (T1), 3 (T2) and 6 (T3) months.

**Total Chol. (mmol∙L^−1^)**	**T1 (*n* = 20)**	**T2 (*n* = 20)**	**T3 (*n* = 18)**
group AB	5.6 ± 0.7	5.6 ± 0.9	5.7 ± 1.1
group BA	5.2 ± 0.8	5.2 ± 1.1	5.4 ± 0.6
**Triglycerides (mmol∙L^−1^)**	**T1 (*n* = 20)**	**T2 (*n* = 20)**	**T3 (*n* = 18)**
group AB	1.9 ± 2.1	1.6 ± 0.7	1.7 ± 0.6
group BA	1.7 ± 1.1	1.6 ± 0.8	1.6 ± 0.6
**Urate (mmol∙L^−1^)**	**T1 (*n* = 20)**	**T2 (*n* = 20)**	**T3 (*n* = 18)**
group AB	0.3 ± 0.1	0.3 ± 0.1	0.3 ± 0.1
group BA	0.4 ± 0.1	0.4 ± 0.1	0.4 ± 0.1

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
