# Peer review of "Oral Supplementation of Specific Collagen Peptides Combined with Calf-Strengthening Exercises Enhances Function and Reduces Pain in Achilles Tendinopathy Patients"

_nutrients, 2019, doi:10.3390/nu11010076_

Reviewer 1 Report

Dear Author,

This paper addresses an interesting question. The paper is very well written, with clear structure and careful explanations throughout. I have just a few small comments on the text, which the authors may wish to address. For example, the changes in tendon microvascularity have to be discussed more critically. Since this is a pilot trial, I recommended to reconsider the design for a RCT, e.g. that the gender distribution is homogenous. In addition, it is argued that proof-of-principle trials of a new drug are more efficiently undertaken using a cross-over design but that subsequent evaluation will require the versatility of trials with a parallel-group design. With a slight re-write this will be a great paper.

 Major issues are commented below. Please find minor comments in the attached PDF-file.  

 Major issues

Study design:

As mentioned by the authors correctly, there is the issue of "carry-over" between treatments, which confounds the estimates of the treatment effects. In practice, "carry-over" effects can be avoided with a sufficiently long "wash-out" period between treatments. However, since determining the exercise therapy was no option, the authors should think about a parallel design for a RCT.

 Returning to running:

Although the protocol demands a recovery period when pain returns during running, the return to running seems to be an interfering factor in respect to the changes in pain and probably microvascularity, since the microvsascularity decreased less, when more participants returned to running but not to their pre-injury level (see 3.3 and 3.4).

Does the general therapeutically approach - Alfredson program- justify returning to running? In that case, the pilot character allows to combine the examination of pain reduction and return to sports. In the discussion you probably should point out, that returning to running with impairments has an influence on the pain reduction and microvascularity

However, in respect to the outcomes the following RCT trial should focus either on pain reduction (without returning to sports) or returning to sport with pain reduction as secondary outcome.

 Outcome:

Since the lipid profile has an influence on inflammatory processes and was examined, the results should be shown.

 Since there is first evidence that the intake of SCP leads to an increased content of elastin and collagen in ligaments and tendons*, a RCT should also focus on tendon stiffness in ultra sound.

 * Schunck, M., Oesser, S. (2013). Specific collagen peptides benefit the biosynthesis of matrix molecules of tendons and ligaments. J. Int. Soc. Sports Nutr. 10, P23.

Author Response

Reviewer 1

 Dear Author,

This paper addresses an interesting question. The paper is very well written, with clear structure and careful explanations throughout. I have just a few small comments on the text, which the authors may wish to address. For example, the changes in tendon microvascularity have to be discussed more critically. Since this is a pilot trial, I recommended to reconsider the design for a RCT, e.g. that the gender distribution is homogenous. In addition, it is argued that proof-of-principle trials of a new drug are more efficiently undertaken using a cross-over design but that subsequent evaluation will require the versatility of trials with a parallel-group design. With a slight re-write this will be a great paper.

Response 

We would like to thank the reviewer for his/her constructive comments and suggestions. Underneath, we have responded to the comments in a point-by-point fashion and revised the text accordingly (see revised Word document).

Major issues are commented below. Please find minor comments in the attached PDF-file.  

Major issues

-Study design:

As mentioned by the authors correctly, there is the issue of "carry-over" between treatments, which confounds the estimates of the treatment effects. In practice, "carry-over" effects can be avoided with a sufficiently long "wash-out" period between treatments. However, since determining the exercise therapy was no option, the authors should think about a parallel design for a RCT.

We agree that for a future RCT, a parallel design is preferred in order to prevent any carry-over effect from the exercise component of a combined treatment. This has been more explicitly added to the Discussion and Conclusion:

P9

So, although the present randomized controlled and double-blinded crossover study found a significant interaction of sCP supplementation on change in VISA-A score during a 6 months calf strengthening plus structured return-to-running program, our findings require duplication in a larger and adequately powered clinical trial with a parallel design.

 P10

Although this pilot study has limited statistical power and requires duplication in a larger randomized clinical trial with a parallel design,…

-Returning to running:

Although the protocol demands a recovery period when pain returns during running, the return to running seems to be an interfering factor in respect to the changes in pain and probably microvascularity, since the microvsascularity decreased less, when more participants returned to running but not to their pre-injury level (see 3.3 and 3.4).

Does the general therapeutically approach - Alfredson program- justify returning to running? In that case, the pilot character allows to combine the examination of pain reduction and return to sports. In the discussion you probably should point out, that returning to running with impairments has an influence on the pain reduction and microvascularity.

However, in respect to the outcomes the following RCT trial should focus either on pain reduction (without returning to sports) or returning to sport with pain reduction as secondary outcome.

Thank you for raising this point. Although from a pathophysiological perspective return-to-running status and tendon microvascularity are likely to interact, the limited statistical power of the present pilot study does not allow us to exclude such a significant interaction using linear mixed modeling. The decision of return-to-running was a function of pain reduction during single leg hopping as well as the successful and pain free achievement of pre-defined milestones during the course of our rehabilitation program. The latter can be justified by the well-established fact that progressive (and sports-specific) tendon loading is required for successful tendon healing and return-to-sport.[31] Furthermore, the milestones described in Appendix 2 entailed significantly lower Achilles tendon loading than present during pre-injury sports participation. Although the VISA-A questionnaires are designed to quantify both ADL and running-associated Achilles tendon pain, we cannot exclude that running with incompletely healed tendon pathology may have introduced a certain level of bias towards the overall VISA-A score. As such, in a future RCT we would recommend to assess both ADL-dependent and running sport-dependent Achilles tendon pain on a visual analog scale as secondary outcome measures.

Revised accordingly in the Discussion P10

-Outcome:

Since the lipid profile has an influence on inflammatory processes and was examined, the results should be shown.

Thank you for this comment. Both in group AB and BA, non-fasting total cholesterol, triglycerides and uric acid remained unchanged during the course of the intervention. As per the reviewer’s suggestion we have created a table to summarize these data and revised accordingly.

Total   Cholesterol (mmol.L-1)

T1

T2

T3

group AB

5.6±0.7

5.6±0.9

5.7±1.1

group BA

5.2±0.8

5.2±1.1

5.4±0.6

Triglycerides   (mmol.L-1)

T1

T2

T3

group AB

1.9±2.1

1.6±0.7

1.7±0.6

group BA

1.7±1.1

1.6±0.8

1.6±0.6

Urate (mmol.L-1)

T1

T2

T3

group AB

0.3±0.1

0.3±0.1

0.3±0.1

group BA

0.4±0.1

0.4±0.1

0.4±0.1

 Since there is first evidence that the intake of SCP leads to an increased content of elastin and collagen in ligaments and tendons*, a RCT should also focus on tendon stiffness in ultra sound.

* Schunck, M., Oesser, S. (2013). Specific collagen peptides benefit the biosynthesis of matrix molecules of tendons and ligaments. J. Int. Soc. Sports Nutr. 10, P23.

Although the study by Schunk and Oesser is an in vitro study on fibroblasts (and obviously did not study the oral intake of sCPs), the 50% increase in elastin content is indeed an interesting observation that warrants further in vivo work. Based on the recent shear wave elastography work by Coombes et al 2018 showing lower Achilles tendon elastic modulus, we agree that it would be interesting to focus on tendon stiffness changes during an RCT.

Revised accordingly in Discussion (last paragraph)

Minor comments

Thank you for the detailed (Minor) comments as mentioned in the pdf.

Please find our response/corrections below:

P2

Citation.

Zdzieblik et al. (11) was included as it contains a table that shows the high glycine content and amino acid composition of hydrolysed collagen, however this citation has now been replaced by Watanabe-Kamiyama et al 2010.

We have added the reference to the study by Dressler et al:

There is also first evidence, that collagen peptides might improve functional ankle properties in chronic ankle instability{Dressler, 2018 #33728}.

-Is there evidence, whether SCP influence the vascularity or do you use the vascularisation as a marker to check this fact?

Then, you probably should point out, that an increased (micro)vascularity is associated with chronic (painful) tendon lesions* and therefore a change in vascularity might be an hinge for improved tendon's architecture.

*Fenwick et al. (2002). The vasculature and its role in the damaged and healing tendon. Arthritis Res. 2002; 4(4): 252–26

Thank you for this suggestion. This is a useful addition to explain more clearly why we aimed to study microvascularity in the context of structural tendon degeneration.

It has been well established that increased (micro)vascularity is associated with chronic (painful) and tendon lesions{Fenwick, 2002 #30003}. As a change in (micro)vascularity might be a hinge for improved tendon architecture, our aim was to check whether specific collagen peptides (sCPs) influence microvascular changes in tendinopathic areas of the Achilles tendon using real-time contrast-enhanced ultrasonography (CEUS)[15]

 -The following investigation refers to an inceased cartillage content and could be included: Mc Alindon et al. (2011). Change in knee osteoarthritis cartilage detected by delayed gadolinium enhanced magnetic resonance imaging following treatment with collagen hydrolysate: a pilot randomized controlled trial. Osteoarthritis Cartilage. 2011 Apr;19(4):399-405.

Thank you for this suggestion. We have added this reference.

-Is there an explanation why the dosage was split?

About 15-30 min after oral ingestion of collagen hydrolysates, free and peptide forms of serum hydroxyproline levels significantly increase and reach a maximum after 30-60 min (Iwai et al 2005). Serum hydroxyproline levels are almost back to baseline level after 7 hours. (Ichikawa et al. 2010) Int. J. Food Sci. Nutr. 61 (1)). Consequently, in order to maximize any potential synergistic effect of orally ingested sCPs with bi-daily calf strengthening exercises (as per  Alfredson’s protocol) on tendon collagen synthesis, our participants were instructed to ingest the sCPs 30 min before each exercise session. Please note that in the study by Shaw et al, 2017, a standard oral gelatin product, rather than hydrolyzed collagen was used. In comparison with orally ingested hydrolysed collagen, peak serum amino acid availability following the oral intake of gelatin is delayed by 30-60 min. (Imoaka et al. Res Comm Chem Path Pharm (1992) 78, 1) Although this was not investigated as a separate outcome measure, the small time window between the bi-daily intake of a sachet with 2.5g of sCPs  and calf strengthening exercise may have contributed to the high compliance rate observed in our trial.

-Why only exclusion of pregnant but not breastfeeding women?

Due to the risk of transplancental migration of microbubbles the use of DEFINITY contrast agent is only contra-indicated in pregnant, but not in breastfeeding women.

-Can you explain why the intake occurred 30 min before exercise program?

To my knowledge, it can be assumed, that the stimulation of anabolic processes might be influenced positively by the administration of SCP immediately before or after the exercise program by virtue of its kinetics - maximum plasma concentration one to two hours after ingestion. First investigations showed an improvement of the collagen metabolism of ligaments by the administration of gelatine before or after physical activity with a superior effect in subject receiving the supplement before the exercise program* *Shaw, G et al. (2017). Vitamin C-enriched gelatin supplementation before intermittent activity augments collagen synthesis. Am. J. Clin. Nutr. 105, 136–143

Please see also explanation above. In order to clarify this further the following paragraph has been added to the Discussion:

In order to maximize any potential synergistic effect of orally ingested sCPs with bi-daily calf strengthening exercises (as per Alfredson’s protocol) on tendon collagen synthesis, our participants were instructed to ingest the sCPs 30 min before each exercise session. About 15-30 min after oral ingestion of collagen hydrolysates, free and peptide forms of serum hydroxyproline levels significantly increase and reach a maximum after 30-60 min and are almost back to baseline level after 7-12 hours[24,34]. Although this was not investigated as a separate outcome measure, the small time window between the bi-daily intake of a sachet with 2.5g of sCPs  and calf strengthening exercise may have contributed to the high compliance rate observed in our trial. A recent proof-of-concept study by Shaw et al (2017) showed a dose-dependent improvement of collagen metabolism in ligaments by the oral administration of gelatine 60 min before jumping exercises[7]. This specific time window was based on the fact that peak availability of free and peptide forms of serum hydroxyproline is respectively 1 and 2 hrs following the oral ingestion of gelatin[35]. Based on the new insight that the stimulation of exercise-induced anabolic processes might be influenced positively by the oral administration of sCPs immediately before an exercise program[7], we recommend that future study designs should take absorption kinetics of the available forms of gelatin or sCPs into consideration.

-Was also a ‘saftey labor’ collected?

Yes, a 500 uL serum sample at T1, T2 and T2 was stored at -80 deg Celsius. However, as no serious adverse events were observed throughout the trial, no further analyses were performed.

Revised accordingly on page 4

In addition, at each time point a 500 uL serum sample was stored at -80 deg Celsius for further analyses in case of a serious adverse event related to the intake of the investigational product.

-pleased add mixed ANOVA

Thank you, this has been added.

-You can refer to fig. 1.

This sentence has can been changed as follows:

Following initial screening, 40 potential candidates (18 men/22 women, age 43±11 yrs) were excluded or withdrew for the reasons mentioned in Figure 1.

-Table 1

Since there is a small number of participants matching is the superior assignment method  However, is there a predisposition of gender? Gender distribution should be taken under consideration in the RCT.

This is a good point and we believe we already mentioned this limitation in the Discussion:

Given the small scale of the study and unequal distribution of men and women in our 2 study groups, there is a potential of selection bias. Although the adaptability of tendon to loading differs in men and women[33], we are not aware of any intervention study that specifically showed clinically relevant gender difference in VISA-A scores following a calf strengthening program. To evaluate homogeneity of the data at baseline between the 2 groups a discriminant analysis was carried out. The data revealed no statistically significant differences between both study groups. Unfortunately, the relatively low number of female participants do not allow for a gender-based inference.

We agree, however that in future RCTs gender should be equally distributed and have added the following sentence to this paragraph:

As women may be more prone to develop Achilles tendinopathy{Wezenbeek, 2018 #33753;Wezenbeek, 2018 #33754}, future RCTs should aim to include an equal number of men and women in both study arms.

-Isn't the lack of symptoms an exclusion criterion? How are 0 month possible?

The 0 months refers to the asymptomatic tendons in the study:

# all tendons, based on pooled data of both symptomatic and asymptomatic tendons.

 -Is this right, since the means of groups are 24 and 18?

Thank you for notifying this error. Underneath, the correct averages (min-max range), which includes also the zeros for the asymptomatic tendons.

Symptom duration (months) &#

51 (0-360)

46 (0-180)

59 (0-360)

 -Could you please adjust the significance levels of the same group at same height

Thank you. This has been adjusted now.

-In respect to the supplementation and following the training protocol for 6 month this really good compliance

Thank you. To emphasize this we have included the following sentence to our Discussion:

Although this was not investigated as a separate outcome measure, the small time window between the bi-daily intake of a sachet with 2.5g of sCPs  and calf strengthening exercise may have contributed to the high compliance rate observed in our trial.

-How did you figure that out?

At both T2 and T3 we asked our participants whether they thought they had been taking the sachets with sCPs or placebo. At the end of the trial and after deblinding this information was matched with true allocation order.

Revised accordingly in the Methods section

-Did anyone of the Tendoforte-then-placebo-group return to "avoid runnung" after stopping the SCP supplementation?

No, none of our participants in the AB group didn’t. Revised accordingly to paragraph 3.3.

-Again: Watanabe-Katakame 2010 would be more appropriate since they measured the SCP composition

Thank you, we have replaced reference 11 by Watanabe-Kamiyama et al 2010 when referring to the sCP composition.

‘So, although the present randomized controlled and double-blinded crossover study found a significant interaction of sCP supplementation on change in VISA-A score during a 6 months calf strengthening plus structured return-to-running program, our findings require duplication in a larger and adequately powered clinical trial.’ Could you please mention this in your results. According to the results there is only an interaction between VISA and vascularity

In paragraph 3.2 of our Results section we have slightly rephrased this important finding as follows:

Importantly, our LMM analyses showed that there was a statistically significant interaction from our sCP randomization order ((LMM, F(2,762)=20.3, p<0.0001)) on change in VISA-A scores indicating a significant difference between the groups in evolution of the VISA-A scores over time in relation to sCP versus placebo supplementatio

Reviewer 2 Report

Title was long and awkward. Please reword. A suggestion is something on the following line: A combination of oral supplementation of specific collagen peptides and exercise enhance function and reduced pain in Achilles tendon patients. Improvement in Achilles tendon pain is not logical. Tendon does not feel pain. Pain perception? 

 Lines 3-8 in the Abstract are detailed methodology that seems wordy and confusing. Can the authors leave these details to the manuscript and simply list them abbreviated? For example: Participants were given a placebo or collagen peptides and bi-daily calf strengthening program for three months then oral supplementation switched to the opposite treatment… something on those lines. But this is a minor point.    

Is the acronym IOC a standard? Would readers of this journal be able to understand from the context?

Page 2. "collagen sysnthesis[7] and improve tendon- and joint-related pain.[9,10]"

Do the authors mean reduce pain? Improving pain is not clear and can mean reduce, abolish completely, etc. I understand that they mean improve the well-being of the patient but this needs rewording.

Methods: I recommend a schematic that explains that the study flow chart. Either add one here or move Figure 1 to this area and refer to it in text.  

It was not clear from the manuscript what T1, T2, T3 were. I understood time points but this was not spelled out. Spell out the first time T1, T2, T3 appear in text. 

Microvascular reduction data seem to an important point that can tip the balance of quality in this manuscript because the rest of data are really just the pain scale using VISA-A. Are there any animal studies that show reduction in microvasculature by histology, CD31 staining? Doppler? CEUS? both? I realize that a biopsy from the patients is not feasible but wanted to see if a discussion of this point can lead us to a better appreciation of what happens the tendon in animals after eccentric training of gastro etc. 

Author Response

We would like to thank Reviewer 2 for his/her comments. We have provided a point-by-point response and also attached the revised version of our manuscript in the attached document which includes revisions based on the valuable comments from both reviewer 1 and 2.

Reviewer 2

Title was long and awkward. Please reword. A suggestion is something on the following line: A combination of oral supplementation of specific collagen peptides and exercise enhance function and reduced pain in Achilles tendon patients. Improvement in Achilles tendon pain is not logical. Tendon does not feel pain. Pain perception? 

Thank you for the suggestion. The title has been revised in accordance with your suggestion

Oral supplementation of specific collagen peptides combined with tailored exercise enhances function and reduces pain in Achilles tendinopathy patients

Lines 3-8 in the Abstract are detailed methodology that seems wordy and confusing. Can the authors leave these details to the manuscript and simply list them abbreviated? For example: Participants were given a placebo or collagen peptides and bi-daily calf strengthening program for three months then oral supplementation switched to the opposite treatment… something on those lines. But this is a minor point.   

Based on the suggestions from Reviewer 1 we have tried to abbreviate and simplify our methodology in the Abstract as follows:

Participants were given a placebo or specific collagen peptides (TENDOFORTE®) in combination with a bi-daily calf strengthening program for 6 months. Group AB received specific collagen peptides for the first 3 months before crossing over to placebo. Group BA received placebo first before crossing over to specific collagen peptides. At baseline, 3 and 6 months, VISA-A questionnaires and microvascularity measurements through contrast-enhanced ultrasound were obtained in 20 patients. 

Is the acronym IOC a standard? Would readers of this journal be able to understand from the context?

Thank you. This has been changed to International Olympic Committee (IOC)

Page 2. "collagen sysnthesis[7] and improve tendon- and joint-related pain.[9,10]"

Do the authors mean reduce pain? Improving pain is not clear and can mean reduce, abolish completely, etc. I understand that they mean improve the well-being of the patient but this needs rewording.

Thank you. We have replaced the word ‘improve’ by ‘reduce’

Methods: I recommend a schematic that explains that the study flow chart. Either add one here or move Figure 1 to this area and refer to it in text.  

This is a good suggestion. Figure 1 has been moved to page 2.

It was not clear from the manuscript what T1, T2, T3 were. I understood time points but this was not spelled out. Spell out the first time T1, T2, T3 appear in text. 

Thank you for notifying this omission. This was indeed only mentioned in the flow diagram and has now also been added in the abstract:

At baseline (T1), 3 (T2) and 6 (T3) months, …

 Microvascular reduction data seem to an important point that can tip the balance of quality in this manuscript because the rest of data are really just the pain scale using VISA-A. Are there any animal studies that show reduction in microvasculature by histology, CD31 staining? Doppler? CEUS? both? I realize that a biopsy from the patients is not feasible but wanted to see if a discussion of this point can lead us to a better appreciation of what happens the tendon in animals after eccentric training of gastro etc. 

Thank you for this suggestion. As far as we know, there are no peer-reviewed mechanistic animal studies that have looked at the microvasculature following (eccentric) tendon loading. There are, however, some other interesting cell-line studies on e.g. the role of endostatin and its role in controlling microvascularity in the collagen matrix. Most recently the work from Wezenbeek et al (AJSM 2018) sheds some new light on the blood flow response in the development of Achilles tendinopathy. In accordance, we have added the following paragraph to our Discussion to put our findings further into perspective:

Eccentric tendon loading has shown to be a safe and effective method to reduce pain and to improve tendon structure[44]. Although PDU-based studies have been equivocal[39-43], eccentric tendon loading has been shown to reduce the number of neovessels in the tendinopathic area of the tendon matrix. The latter is considered to be an important etiological mechanism for the beneficial outcome of a calf strengthening program as also applied in the current study[45]. Cell-line studies have shown that tenocytes produce the antiangiogenic factor endostatin, a proteolytic fragment of Collagen XVIII, in response to physiological mechanical load, thus limiting neo-angiogenesis[46]. Although endostatin response to eccentric loading has not been investigated in diseased tendons, several mechanistic studies[46-48] indicate that only the right amount of tissue loading increases endostatin expression and as such may reduce microvascularity as observed in the present study. As the collagen matrix of tendon is a highly mechanosensitive tissue, cytokine homeostasis and cell survival underlie an intimate balance between adequate biomechanical stimuli and disturbance through load deprivation and overload. This delicate balance between tendon blood flow and loading pattern was recently highlighted in a study by showing that the risk of developing Achilles tendinopathy increased if blood flow increase in the tendon after running was reduced[35]. Based on these new insights, future studies on novel therapeutic strategies for tendinopathy should preferably assess tendon blood flow response both at rest as well as following a standardized tendon loading protocol.
